# Gait outcomes of older adults receiving subacute hospital rehabilitation following orthopaedic trauma: a longitudinal cohort study

Saira A Mathew,[1,2] Paul Varghese,[3,4] Suzanne S Kuys,[5] Kristiann C Heesch,[1] Steven M McPhail[1,2]

► Prepublication history and additional material are available. To view these files, please visit the journal online (http://dx.doi.org/10.1136/bmjopen-2017-016628).

[1]School of Public Health and Social Work and Institute of Health and Biomedical Innovation, Queensland University of Technology, Brisbane, Australia
[2]Centre for Functioning and Health Research, Metro South Health, Brisbane, Australia
[3]School of Medicine, The University of Queensland, Queensland, Australia
[4]Geriatric Assessment and Rehabilitation Unit, Princess Alexandra Hospital, Brisbane, Australia
[5]School of Physiotherapy, Australian Catholic University, Brisbane, Australia

**Correspondence to**
Dr Steven M McPhail; steven.mcphail@qut.edu.au

## ABSTRACT

**Objectives** This study aimed to describe gait speed at admission and discharge from inpatient hospital rehabilitation among older adults recovering from orthopaedic trauma and factors associated with gait speed performance and discharge destination.

**Design** A longitudinal cohort study was conducted.

**Setting** Australian tertiary hospital subacute rehabilitation wards.

**Participants** Patients aged ≥60 years recovering from orthopaedic trauma (n=746, 71% female) were eligible for inclusion.

**Interventions** Usual care (multidisciplinary inpatient hospital rehabilitation).

**Primary and secondary outcome measures** Gait speed was assessed using the timed 10 m walk test. The proportion of patients exceeding a minimum gait speed threshold indicator (a priori 0.8 m/s) of community ambulation ability was calculated. Generalised linear models were used to examine associations between patient and clinical factors with gait speed performance and being discharged to a residential aged care facility.

**Results** At discharge, 18% of patients (n=135) exceeded the 0.8 m/s threshold indicator for community ambulation ability. Faster gait speed at discharge was found to be associated with being male (B=0.43, 95% CI −0.01 to 0.87), admitted with pelvic (B=0.76, 95% CI 0.14 to 1.37) or multiple fractures (B=1.13, 95% CI 0.25 to 2.01) (vs hip fracture), using no mobility aids (B=−0.93, 95% CI −1.89 to 0.01) and walking at a faster gait speed at admission (B=5.77, 95% CI 5.03 to 6.50). Factors associated with being discharged to residential aged care included older age (OR 1.06, 95% CI 1.03 to 1.10), longer length of stay (OR 1.01, 95% CI 1.01 to 1.02), having an upper limb fracture (vs hip fracture) (OR 2.81, 95% CI 1.32 to 5.97) and lower Functional Independence Measure cognitive score (OR 0.89, 95% CI 0.86 to 0.92).

**Conclusions** Patients with a range of injury types, not only those presenting to hospital with hip fractures, are being discharged with slow gait speeds that are indicative of limited functional mobility and a high risk of further adverse health events.

## Strengths and limitations of this study

► The study examined discharge gait speeds and discharge destinations of older adults admitted for subacute inpatient hospital rehabilitation for a range of orthopaedic injury types.
► The inclusion of all admissions meeting prespecified inclusion criteria reduced the risk of sampling bias.
► This study was conducted at a single subacute hospital rehabilitation unit and findings may not be generalised to dissimilar clinical settings or healthcare systems.

## INTRODUCTION

Orthopaedic injuries associated with age-related frailty carry a substantial health and quality-of-life burden for older adults and consume considerable healthcare resources.[1 2] Older adults who sustain orthopaedic trauma may experience difficulty returning to premorbid functional status and participating in activities of daily living.[3] The risk of disability, postsurgical complications and subsequent fractures increases with older age.[4] Older adults who have been hospitalised with a fragility fracture may require additional supports or admission to a residential aged care facility.[3]

To reduce the risk of requiring admission to an aged care facility and to optimise functional recovery, many patients receive inpatient hospital rehabilitation.[5] In a hospital rehabilitation setting, mobility is frequently evaluated through performance-based measures such as gait speed assessment.[6 7] Gait ability is a key rehabilitation outcome, a determinant of discharge destination, and limitations in gait may persist at discharge from subacute rehabilitation.[8 9] Slow gait speed has been associated with disability risk, hospitalisation and premature mortality, and improvements in gait speed have been associated with lower risk of mortality and

independence in community ambulation.[6 7 10 11] An ability to walk 46m at a speed between 0.9 and 1.2 m/s has been reported to be the minimum requirement for older adults to ambulate independently in the community.[12] However, some researchers propose that a gait speed of 0.8 m/s is the minimum required to ambulate independently and participate in community activities.[12 13] Gait speeds of 0.4–0.8 m/s have been associated with independently performing activities of daily living, but with limited community ambulation.[13 14] Individuals with gait speeds <0.4 m/s are likely to require assistance with performing some household activities and require full assistance (eg, wheelchair use) for community activities.

Previous studies have reported estimates of clinically important differences in gait speed during the first year of recovery from a fracture and the characteristics of independent community ambulators.[7 15 16] Small, but meaningful, improvements in gait speed, 0.10–0.17 m/s, have been observed during recovery from hip fractures, and substantial improvements, 0.17–0.26 m/s, have also been observed.[16] While proficient motor control and sound cognitive function have been associated with independent community ambulation, few studies have reported factors associated with changes in gait speed and discharge destination among older adults with orthopaedic trauma related injuries.[7 15 16]

This study had three objectives to be investigated in a sample of older adults admitted to a subacute hospital rehabilitation unit with orthopaedic trauma related injuries. The first was to describe gait performance at admission and discharge from hospital rehabilitation. The second was to calculate the proportion of the sample who met a minimum gait speed threshold indicator for community ambulation at discharge and MCID (minimal clinically important difference) for change in gait speed during inpatient hospital rehabilitation. The third objective was to examine which patient and clinical factors were associated with (a) change in gait speed, (b) gait speed at discharge and (c) discharge to a residential care facility (interim care, hostel or residential aged care facilities).

## METHODS
### Design
A longitudinal cohort study with two assessment points was conducted.

### Participants and settings
Participants were patients, aged ≥60 years, who presented to a tertiary hospital facility with fracture-related injuries that required an inpatient admission. At the participating hospital, patients typically receive acute management of their fracture (eg, reduction and stabilisation) and acute inpatient care in a specialised orthopaedics unit. Frail older adults who are not able to safely return home following their inpatient admission may receive further subacute inpatient rehabilitation within the hospital. This participating subacute hospital rehabilitation unit

was a 76-bed facility providing comprehensive geriatric medicine and multidisciplinary therapy. This included weekday rehabilitative therapies (eg, physical therapy), nurse-supervised mobility (eg, walking from the bedside to a ward-based dining area for meals for patients with appropriate mobility) and weekly case conferencing within the multidisciplinary team.

Patients were not eligible for inclusion if they were admitted for non-orthopaedic traumatic injuries or for injuries that included serious neurological insults (eg, spinal cord injury) or if they were transferred to another hospital without completing subacute inpatient hospital rehabilitation at the participating facility. Patients who died during their inpatient stay were also excluded.

### Procedures
Patients admitted for subacute inpatient hospital rehabilitation at the participating hospital routinely complete assessments within 24 hours of their admission to the rehabilitation unit or prior to their discharge from the rehabilitation unit. However, on occasions of admission or discharge on weekend days, these assessments may be completed up to 72 hours of admission or prior to discharge. As part of these two assessments, gait performance was assessed by their treating physiotherapists using the timed 10 m walk test.[17–19]

### Gait speed assessment
The timed 10 m walk test was used to assess patients' walking ability over 14 m, with the middle 10 m timed using a stopwatch. In the present study, treating physiotherapists followed standardised testing procedures,[17–19] including instructions to walk at a comfortable and safe pace and permitting the use of a mobility aid (eg, a walking frame). Patients unable to complete the test without physical assistance of another person were considered unable to complete the test.

### Clinical and patient characteristics
Clinical and patient characteristics collected at admission included age, sex and type of injury(s). At admission and discharge from the subacute inpatient rehabilitation unit, the use of a mobility aid was recorded, and two functional measures were completed: the Functional Independence Measure (FIM motor and FIM cognitive) by an occupational therapist trained in the FIM assessment procedures and the Modified Elderly Mobility Scale by a physiotherapist.[19 20] At discharge, the length of stay in the hospital rehabilitation unit was recorded. Three rehabilitation outcomes were of interest for this study: change in gait speed (between the admission and discharge assessment), gait speed at discharge and discharge to a residential aged care facility.

### Statistical analysis
For the purpose of describing gait performance, participants were classified into one of four categories: (a) unable to complete the timed 10 m walk test at admission and discharge; (b) unable to complete the timed 10 m walk test at admission, but able at discharge; (c) able to complete the test at admission and discharge;

(d) able to complete the test at admission, but unable at discharge. Frequency histograms illustrate the distribution of discharge gait speed and change in gait speed after excluding patients unable to complete the assessments at admission and discharge. Frequencies (and proportion) of patients who met the previously reported minimum threshold gait speed of 0.8 m/s,[18] which is indicative of ability to successfully ambulate in the community, were calculated. Also calculated were the frequencies and proportions of patients who experienced a change in gait speed that exceeded a previously reported estimate of MCID of 0.10 m/s.[21]

Three generalised linear models were used to examine patient and clinical characteristics associated with (a) change in gait speed, (b) discharge gait speed and (c) whether the patient was discharged to a residential aged care facility (yes or no). The Gaussian family and identity link option were used for both gait speed models, while the binomial family and logit link option were used for the model examining discharge to a residential aged care facility. Patient and clinical characteristics included as independent variables in all models were patient age (continuous, in years), sex, length of stay (continuous, in days), type of injury (nine categories), admission gait speed (continuous, in m/s) and use of a mobility aid at admission (no/yes). In the discharge destination model, three additional variables were included: admission Modified Elderly Modified Scale score, admission FIM cognitive score and admission FIM motor score.

Fifty-nine patients (7.9%) had missing data on at least one variable, with missing data considered likely to be missing at random. Multiple imputations by chained equations were used to enable inclusion of these patients in the modelling.[22] Multicollinearity was assessed using variance inflation factors and tolerance statistics. All were <5, indicating multicollinearity was not present. Robust standard errors (Huber-White Sandwich Variance Estimator approach) were used to account for potential heteroscedasticity.[23 24]

For the aforementioned primary generalised linear models that examined (a) change in gait speed and (b) discharge gait speed, participants unable to complete the time 10 m walk test were assigned a gait speed of 0 m/s. Sensitivity analyses were also conducted for these two generalised linear models. These sensitivity analyses excluded patients assigned a gait speed of 0 m/s for the primary analyses due to being unable to complete the timed 10 m walk test at both admission and discharge assessments. This was conducted to examine whether the exclusion of this subset of patients influenced the findings of the aforementioned linear models. All analyses were conducted using Stata V.13 (StataCorp) and alpha was set at a 0.05 significance level.

## RESULTS

A total of 774 patients were initially eligible for inclusion (figure 1). Of these, 28 were not included: 22 (2.8%) patients were transferred to another hospital unit prior to assessment and six (0.8%) died. The characteristics of the remaining 746 patients (96.4%) are presented in table 1. A summary of gait speed at each assessment and improvement in gait speed has been provided in table 2 and online supplementary figure S1. In short, most (76.4%) patients who were able to complete the test at both assessments improved their gait speed by more than MCID in gait of 0.10 m/s,[21 25] while just 1.3% decreased by a clinically meaningful margin.

### Factors associated with gait speed at discharge from hospital rehabilitation

Patients with pelvic fractures (p<0.01) and multiple fractures (p<0.01) had faster gait speed at discharge than patients with neck of femur fractures (table 3). Compared with female patients, male patients were more likely to have faster gait speed at discharge (p=0.05). Also, patients who used a mobility aid at admission were more likely to have walked slower at discharge compared with those who did not use a mobility aid (p=0.05).

### Factors associated with change in gait speed during inpatient hospital rehabilitation

Patients with pelvic fractures (p<0.01) and multiple fractures (p<0.01) demonstrated greater improvement in

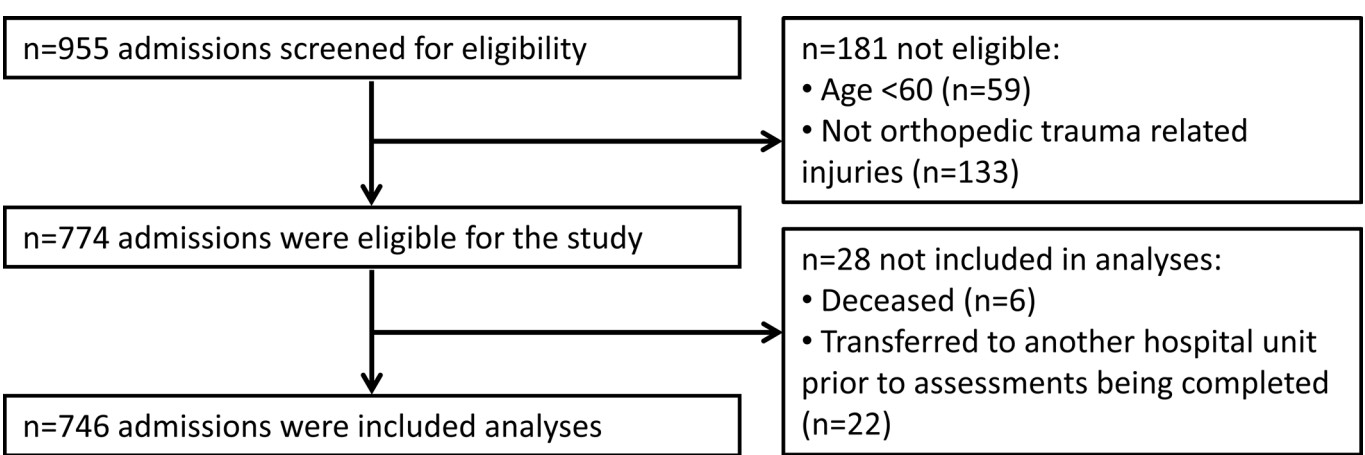

**Figure 1** Participant flow diagram.

**Table 1**  Patient and clinical characteristics

| Characteristics | | |
|---|---|---|
| Age (mean, SD) | 82 | 8 |
| Gender (n, %) | | |
| Male | 211 | 28.3 |
| Female | 535 | 71.7 |
| Type of injury (n, %) | | |
| Neck of femur | 403 | 54.0 |
| Pelvic | 86 | 11.5 |
| Upper limb | 63 | 8.4 |
| Spinal | 56 | 7.5 |
| Femoral | 41 | 5.5 |
| Multiple | 38 | 5.1 |
| Other axial | 31 | 4.2 |
| Lower limb | 28 | 3.8 |
| Mobility aid at admission assessment (n, %) | | |
| Walking frame (four-wheeled walker or hopper) | 611 | 82 |
| Stick (single, two single, four-point) | 49 | 6.6 |
| Wheelchair | 39 | 5.2 |
| Nil | 37 | 4.9 |
| Others (crutches) | 10 | 1.3 |
| Discharge destination (n, %) | | |
| Home | 614 | 82.3 |
| Residential aged care facility (high-level care facility) | 78 | 10.4 |
| Hostel (low-level care facility) | 32 | 4.3 |
| Interim | 22 | 3.0 |
| Length of stay in days (median, IQR) | 35 | 23–56 |
| Mobility aid at discharge assessment (n,%) | | |
| Walking frame (four-wheeled walker or hopper) | 448 | 60.2 |
| Stick (single, two single, four-point) | 207 | 27.7 |
| Nil | 62 | 8.3 |
| Wheelchair | 19 | 2.5 |
| Others (crutches) | 10 | 1.3 |

**Table 2**  Mean gait speed at admission and discharge

| Ability to complete timed 10 m walk at each assessment | n | Mean gait speed at admission (SD) | Mean gait speed at discharge (SD) |
|---|---|---|---|
| Unable at admission and discharge | 61 | 0 (–) | 0 (–) |
| Unable at admission and able at discharge | 303 | 0 (–) | 0.53 (0.22) |
| Able at admission and unable at discharge | 1 | 0.18 (–) | 0 (–) |
| Able at admission and discharge | 381 | 0.37 (0.21) | 0.64 (0.25) |

Dash (–) indicates SD not applicable as there was no variation in gait speed for that category.

than those with neck of femur fractures. A lower FIM cognitive score was associated with a greater likelihood of being discharged to a residential aged care facility (p<0.001).

### Sensitivity analyses

The sensitivity analysis findings were consistent with those of the primary analyses of factors associated with discharge gait speed and change in gait speed (online supplementary table S1). The coefficient point estimates were similar to those in the primary analyses, and discrepancies between the primary and sensitivity analysis confidence intervals were of a small magnitude.

### DISCUSSION

This study is the first to examine gait speeds at discharge from subacute hospital rehabilitation specifically in patients with orthopaedic trauma related injuries as well as to examine associations between patient and clinical characteristics with gait outcomes and discharge to residential aged care. The gait outcomes ought to be interpreted in the context of the high level of frailty present in the patient sample. The findings could be interpreted as positive with the majority (91.7%) of patients able to walk at least a small distance without assistance of another person at discharge, a commendable achievement given that approximately half of the participants were unable to walk without assistance of another person at admission to rehabilitation. Furthermore, a (relatively) low proportion (10.4%) of participants was discharged to a residential aged care facility for ongoing high-level nursing care. However, this study also highlighted that only 18% of participants achieved the previously reported 0.8 m/s threshold indicator for ability to ambulate in community environments.[18]

Previous reports have proposed that gait speeds above 1.0 m/s are desirable and those lower than 0.5 m/s are associated with poor function and health.[26–28] The slow gait speed observed at completion of inpatient rehabilitation in the present study is not an indictment of the

gait speed than patients with neck of femur fractures (table 3). Male gender (p=0.05) was associated with greater improvement in gait speed, while use of a mobility aid at admission was associated with less improvement in gait speed (p=0.05).

### Factors associated with discharge to a residential aged care facility

Older age (p<0.001) and prolonged length of stay (p<0.001) were associated with a lower likelihood of being discharged home (table 4). Participants with upper limb fractures were less likely to be discharged home (p<0.01)

**Table 3** Summary of coefficients from two generalised linear models examining the association of patient and clinical factors with change in gait speed and gait speed at discharge

| Model-dependent variable | Independent variables | Coefficient×10 | 95% CI | p Value |
|---|---|---|---|---|
| Discharge gait speed (Wald χ2(12)=362.90, p<0.001) | Age | −0.01 | −0.04 to 0.05* | 0.13 |
| | Male gender | 0.43 | −0.01* to 0.87 | 0.05 |
| | Length of stay | −0.04* | −0.01 to 0.03* | 0.26 |
| | Type of injury | | | |
| | Neck of femur | Referent | | |
| | Spinal | 0.09 | −0.43 to 0.62 | 0.73 |
| | Pelvic | 0.76 | 0.14 to 1.37 | <0.01 |
| | Upper limb | −0.57 | −1.18 to 0.02 | 0.06 |
| | Multiple | 1.13 | 0.25 to 2.01 | <0.01 |
| | Femoral | −0.29 | −1.06 to 0.47 | 0.45 |
| | Lower limb | 0.68 | −0.31 to 1.69 | 0.17 |
| | Other axial | 0.44 | −0.46 to 1.35 | 0.33 |
| | Gait speed* | 5.77 | 5.03 to 6.50 | <0.001 |
| | Use of mobility aid* | −0.93 | −1.89 to 0.01 | 0.05 |
| Change in gait speed (Wald χ2(12)= 208.81, p<0.001) | Age | −0.01 | −0.04 to 0.05* | 0.13 |
| | Male gender | 0.43 | −0.01* to 0.87 | 0.05 |
| | Length of stay | −0.04* | −0.01 to 0.03 | 0.25 |
| | Type of injury | | | |
| | Neck of femur | Referent | | |
| | Spinal | 0.09 | −0.43 to 0.62 | 0.73 |
| | Pelvic | 0.76 | 0.14 to 1.37 | <0.01 |
| | Upper limb | −0.57 | −1.18 to 0.02 | 0.06 |
| | Multiple | 1.13 | 0.25 to 2.01 | <0.01 |
| | Femoral | −0.29 | −1.06 to 0.47 | 0.45 |
| | Lower limb | 0.68 | −0.31 to 1.69 | 0.17 |
| | Other axial | 0.44 | −0.46 to 1.35 | 0.33 |
| | Gait speed* | −4.22 | −4.96 to −3.49 | <0.001 |
| | Use of mobility aid* | −0.93 | −1.89 to 0.01 | 0.05 |

*Coefficients have been multiplied by 100.

inpatient model of care at the participating facility. It was encouraging that over three quarters of participants could be considered to have improved their gait speed by a clinically meaningful margin during their stay. The mean gait speed at discharge was consistent with a systematic review of prior reports from similar subacute hospital settings that included non-orthopaedic patients.[29] However, the findings highlight the importance of planning for an integrated transition from hospital to community-based services in the initial posthospitalisation period for older people recovering from orthopaedic trauma. This may include engagement with community-based therapies for patients who have potential to extend gains attained during inpatient rehabilitation, as well as supportive care interventions that seek to maximise patients' health-related quality of life and mitigate risk of adverse events. Prior research has highlighted the potential effectiveness

(and cost-effectiveness) of community-based interventions for frail older adults recovering from injury and illness.[30 31]

Prior studies in the field have typically focused on patients recovering from hip fractures and have often emphasised risk of undesirable outcomes following hip fractures.[32–34] The inclusion of patients recovering from fractures affecting a range of body regions in the present study permitted novel contributions, particularly in the analytical models that examined factors associated with outcomes at discharge from inpatient rehabilitation. For example, patients recovering from upper limb injuries were more likely to be discharged to residential aged care facilities than patients recovering from hip fractures. At first, this finding may seem paradoxical given the extent of prior reporting of challenges and functional deficits faced by people recovering from hip fractures,[35 36] but

**Table 4** Summary of ORs from a generalised linear model examining the association of patient and clinical factors with discharge to a residential aged care facility

| Model-dependent variable | Independent variables | OR | 95% CI | p Value |
|---|---|---|---|---|
| Discharge to a residential aged care facility (Wald $\chi 2(15)=7.84$, p<0.001) | Age | 1.06 | 1.03 to 1.10 | <0.001 |
| | Male gender | 0.98 | 0.58 to 1.67 | 0.95 |
| | Length of stay | 1.01 | 1.01 to 1.02 | <0.001 |
| | Type of injury | | | |
| | Neck of femur | Referent | | |
| | Spinal | 1.28 | 0.49 to 3.34 | 0.60 |
| | Pelvic | 0.80 | 0.36 to 1.76 | 0.59 |
| | Upper limb | 2.81 | 1.32 to 5.97 | <0.01 |
| | Multiple | 0.20 | 0.02 to 1.55 | 0.12 |
| | Femoral | 2.02 | 0.85 to 4.82 | 0.11 |
| | Lower limb | 0.83 | 0.22 to 3.10 | 0.79 |
| | Other axial | 0.25 | 0.03 to 2.12 | 0.20 |
| | MEMS score* | 0.97 | 0.89 to 1.05 | 0.46 |
| | FIM cognitive score* | 0.89 | 0.86 to 0.92 | <0.001 |
| | FIM motor score* | 0.98 | 0.96 to 1.00 | 0.15 |
| | Gait speed* | 0.50 | 0.09 to 2.60 | 0.41 |
| | Use of mobility aid* | 0.85 | 0.31 to 2.31 | 0.75 |

*Functional measures assessed at admission.
FIM, Functional Independence Measure; MEMS, Modified Elderly Modified Scale.

it may be explained by the likely high level of frailty in patients requiring inpatient hospital rehabilitation following an upper limb fracture. It is likely that most patients who present to hospital with an upper limb fracture will be discharged soon after their fracture has been stabilised, and relatively few will require subacute inpatient rehabilitation. Of all older adults who sustain an upper limb injury, it is likely to be the most frail, complex or socially vulnerable who will require subacute inpatient hospital rehabilitation.[37] In comparison, a high proportion of patients who sustain a hip fracture may require subacute hospital rehabilitation regardless of their social circumstances or premorbid functioning to continue rehabilitative therapies in an effort to attain independence with mobility tasks prior to discharge from hospital.

Understanding factors associated with patients experiencing the greatest (or least) improvement in gait speed between admission and discharge offers insights useful for clinical teams, health service policy related to inpatient rehabilitation and future research in the field. In the present study, the greatest improvements in gait speed were observed among patients with pelvic fractures or with multiple traumatic injuries, which was

an encouraging finding as patients from these two diagnostic groups have received limited attention in prior research.[29 38] It was also encouraging that patients with a slow baseline gait speed were among those to demonstrate the greatest improvement in gait speed during their inpatient rehabilitation, perhaps indicative of the fulfilment of (at least theoretical) potential for improvement that accompanies starting from a lower base than others in the cohort. However, not requiring the use of a mobility aid at admission was associated with greater improvement in gait speed. It is plausible that patients not requiring a mobility aid at admission to the rehabilitation unit had lower levels of premorbid physical frailty and were more readily able to return to higher gait speeds by discharge than peers in their cohort.

### Strengths and limitations
A strength of this study was the inclusion of a range of orthopaedic injury types present among older adults who were receiving hospital rehabilitation. Second, the inclusion of consecutive admissions meeting the inclusion criteria, with a relatively low rate of missing assessments, reduced the risk of sampling bias. One limitation of this approach was that clinical variables were limited to those recorded for all participants during their routine clinical assessment. This meant that height was not available as a potential covariate for adjustment in the generalised linear models. Another important limitation of the study was that it was conducted at a single geographical location in an industrialised nation in a subacute hospital rehabilitation setting. It is possible that older adults recovering from orthopaedic injuries in dissimilar societies or healthcare systems may not have had comparable gait performances as patients in the present sample. The study design was appropriate for addressing the intended aims, but it did not allow any conclusions to be drawn regarding the effectiveness (or otherwise) of the model of care received by patients in improving gait speed in comparison to other models of care. This topic was considered beyond the scope of the present study, but future investigation of the optimal type and intensity of rehabilitation interventions for improving gait among older adults recovering orthopaedic injuries remains a priority for research.

### CONCLUSION
Most patients were able to complete the 10 m walk test without assistance of another person at discharge, but few were likely to be able to ambulate in community environments. The greatest improvements in gait speed were observed among patients with pelvic fractures and those with multiple traumatic injuries. Also noteworthy was that patients receiving hospital-based rehabilitation following injuries affecting upper limbs were more likely to be discharged to residential aged care facilities than patients recovering from hip fractures. Overall, the findings highlight the importance of planning for an integrated

transition from hospital to community-based services in the initial post-hospital rehabilitation period. This may include engagement with community-based rehabilitative therapies or supportive care interventions to improve health-related quality of life and mitigate risk of further adverse health events.

**Contributors** SSK, PV and SMM: obtaining funding and acquisition of data. SAM, SMM and KCH: study design, analysis and interpretation of data. SAM and SMM: drafting of the manuscript. All authors: interpretation of findings, revision of manuscript and approval of final version.

**Funding** The study was supported in part by the Health Practitioner Research Support Scheme (Queensland Department of Health). SMM is supported by a National Health and Medical Research Council (of Australia) fellowship. The Department of Health Queensland financially supported the study but had no role in the research design, reporting or publishing decision.

**Competing interests** None declared.

**Patient consent** Detail has been removed from this case description/these case descriptions to ensure anonymity. The editors and reviewers have seen the detailed information available and are satisfied that the information backs up the case the authors are making.

**Ethics approval** Metro South Human Research Ethics Committee.

**Provenance and peer review** Not commissioned; externally peer reviewed.

**Data sharing statement** No additional data are available for public dissemination, although further details on statistical analysis are available from the corresponding author on request.

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
