## [Reviewer comments · BMJ Open]

ARTICLE DETAILS

TITLE (PROVISIONAL)	Gait outcomes of older adults receiving subacute hospital rehabilitation following orthopedic trauma: a longitudinal cohort study
AUTHORS	Mathew, Saira; Varghese, Paul; Kuys, Suzanne; Heesch, Kristiann; McPhail, Steven

VERSION 1 - REVIEW

REVIEWER	Richard W Bohannon Campbell University, USA
REVIEW RETURNED	07-Mar-2017

GENERAL COMMENTS	1. Some sentences with a series of points could be written in a more instructive and parallel fashion. For example: "Factors associated with faster gait speed were being male, having non-hip fractures, using no mobility aid, and walking faster on admission." 2. The participant flow chart strikes me as not quite right. The final text box should be the 381 able to perform the assessment at admission and discharge. Others were excluded for the state reasons. 3. Information is provided regarding when assessments were completed. However, these need to be clearly tied to "admission" and "discharge." Perhaps something like "admission assessments of gait speed were completed within 72 hours of admission, on average ??? days after." 4. A histogram cannot examine. They can illustrate. 5. Label column 1 of Table 2. 6. The study may be the first to examine gait speed changes for a specific clinical group at discharge from a subacute setting, but it is not the first study focused on gait speed in such a setting. See: 1) Barthuly AM, Bohannon RW, Gorack W. Gait speed is a responsive measure of physical performance for patients undergoing short-term rehabilitation. Gait Posture 2012;36:61-64. 2) Barthuly AM, Bohannon RW, Gorack W. Limitations in gait speed persist at discharge from subacute rehabilitation. J Phys Ther Sci 2013;25:891-893. 7. There are errors in the presentation of References. Some Journal titles are not italicized. Some article titles include inappropriate capitalization
--

REVIEWER	Tom Overend School of Physical Therapy, Western University London, Ontario, Canada
REVIEW RETURNED	26-Apr-2017

GENERAL COMMENTS	Thanks for the opportunity to review your manuscript. You have looked at gait speed in older adults with various rehabs after orthopedic trauma. You found that only 18% of them got back to at least 0.8 m/sec gait speed. Interesting to read and a well written paper, I just have a few comments.  1. There is no info on ethical permission in the Methods. Was this not required? Did you not have to get their permission to use their data? 2. Table 1- under Discharge destination, you have Interim but there is no data associated with this? Do you need to keep Interim in the table, or did you just forget to add the number and percentage? 3. For other tables you have a variety of decimal points. I am not sure about this journal but suggest that a common format be used (either 2 or 3 decimals). 4. References - be sure to italicize all your journal names. Put a complete page reference for ref #18. Add spaces in ref #1 on second line.
--

VERSION 1 – AUTHOR RESPONSE

Reviewer 1:

1. Some sentences with a series of points could be written in a more instructive and parallel fashion. For example: “Factors associated with faster gait speed were being male, having non-hip fractures, using no mobility aid, and walking faster on admission.”

Response:

We appreciate our attention being drawn to this and in response have made amendments to several sections of text including both the abstract (Page 2, results), as well as the main text of the document where words have been removed to make parallel construction. This is notable on page 8 line 117 and page 11 line 173.

2. The participant flow chart strikes me as not quite right. The final text box should be the 381 able to perform the assessment at admission and discharge. Others were excluded for the state reasons.

Response:

We see how the last box on the figure that the reviewer is referring to here was confusing. The details provided in the last box were not actually referring to patients excluded from the study, but rather gait ability pertaining to completion of assessments. However, by including this in the figure, we have inadvertently given the impression some patients were excluded from the study on the basis of gait performance. This was not the case. In fact, all information that was reported in that bottom part of the participant flow figure was already reported more clearly in Table 2, and we have now more clearly directed the reader to that on line 150-151. To avoid confusion (and duplicating information in Table 2), we have removed the bottom box of the original figure entirely.

3. Information is provided regarding when assessments were completed. However, these need to be clearly tied to “admission” and “discharge.” Perhaps something like “admission assessments of gait speed were completed within 72 hours of admission, on average ??? days after.”

Response:

We have now added further description regarding the timing of assessment and ensured it clearly ties them to admission or discharge (line 79-83). In summary, assessments are typically completed within 24 hours of their admission to the rehabilitation unit or 24 hours prior to their discharge from the

rehabilitation unit. However, on occasions of admission or discharge on weekend days, these assessments are occasionally completed up to 72 hours of admission to the rehabilitation unit or 72 hours prior to discharge. This has now been described more clearly in the text. Although we checked each admission for eligibility, we did not specifically record the hour that the assessment was completed in relation to the hour of admission (and therefore, from our existing data, we cannot report the average number of hours between admission and admission gait assessment etc.). However, in light of the clarification that we have now made in the text regarding assessments typically being completed within one day of admission, I am not sure the specific hour is of much consequence in light of the duration of inpatient length of stay (median 35 days) in the rehabilitation unit. We trust that this additional description has addressed the sentiments of the reviewer in ensuring that the assessments are clearly tied to admission and discharge.

4. A histogram cannot examine. They can illustrate.

Response:

We appreciate this wording error being brought to our attention and have now made this correction (and replaced the text 'examine' with 'illustrate'). (Page 7, line 108)

5. Label column 1 of Table 2.

Response:

We have now labelled column 1 of Table 2 as "Gait performance" (Table 2, page 10)

6. The study may be the first to examine gait speed changes for a specific clinical group at discharge from a subacute setting, but it is not the first study focused on gait speed in such a setting. See: 1) Barthuly AM, Bohannon RW, Gorack W. Gait speed is a responsive measure of physical performance for patients undergoing short-term rehabilitation. *Gait Posture* 2012;36:61-64. 2) Barthuly AM, Bohannon RW, Gorack W. Limitations in gait speed persist at discharge from subacute rehabilitation. *J Phys Ther Sci* 2013;25:891-893.

Response:

We agree, and have now made reference to these studies in the manuscript (e.g., Page 4, line 23)

7. There are errors in the presentation of References. Some Journal titles are not italicized. Some article titles include inappropriate capitalization

Response:

Great, we appreciate this being brought to our attention. We have now made these amendments in keeping with journal guidelines.

Reviewer 2:

Thanks for the opportunity to review your manuscript. You have looked at gait speed in older adults with various rehabs after orthopedic trauma. You found that only 18% of them got back to at least 0.8 m/sec gait speed. Interesting to read and a well written paper, I just have a few comments.

1. There is no info on ethical permission in the Methods. Was this not required? Did you not have to get their permission to use their data?

Response:

Yes, this study did require ethical approval. Although we would often put this information in the methods section, we had tried to follow the journal's instructions for authors and included the

information about ethics and consent in a statements section with the other required statements on Page 18. We are also happy to put it in the methods section at the editor's discretion, we don't mind where it goes, but I think the key thing to say is: yes, the study was indeed carried out in accordance with the requirements of the relevant accredited institutional human research committee etc.

2. Table 1- under Discharge destination, you have Interim but there is no data associated with this? Do you need to keep Interim in the table, or did you just forget to add the number and percentage?

Response

We really appreciate this being drawn to our attention. There was indeed a formatting error that resulted in the Interim care data being located on the wrong line (which gave the impression it was missing). We have now corrected this error (Page 10).

3. For other tables you have a variety of decimal points. I am not sure about this journal but suggest that a common format be used (either 2 or 3 decimals).

Response

Again, we appreciate this oversight being drawn to our attention. We have now amended decimal points to be consistent in all tables.

4. References - be sure to italicize all your journal names. Put a complete page reference for ref #18. Add spaces in ref #1 on second line.

Great, we appreciate this being brought to our attention. We have now made these amendments in keeping with journal guidelines (Page 19).